# GROTHENDIECK GRAPH NEURAL NETWORKS FRAMEWORK: AN ALGEBRAIC PLATFORM FOR CRAFTING TOPOLOGY-AWARE GNNS

## ABSTRACT

Graph Neural Networks (GNNs) typically rely on neighborhoods as the foundation of message passing. While simple and effective, neighborhoods limit expressivity, often no stronger than the Weisfeiler–Lehman (WL) test. We propose the Grothendieck Graph Neural Networks (GGNN) framework, an algebraic platform that generalizes neighborhoods into covers, offering flexible alternatives for defining message-passing strategies. GGNN translates covers into matrices, similar to how adjacency matrices encode neighborhoods, enabling both theoretical analysis and practical implementation. Within this framework, we introduce the cover of sieves, inspired by category theory, which captures rich topological features. Based on this cover, we design Sieve Neural Networks (SNN), which produce the matrix form of the cover of sieves, generalizing the adjacency matrix. Experiments show that SNN achieves zero failures on graph isomorphism tasks (SRG, CSL, BREC) and improves topology-aware evaluation via a label propagation probe. These results demonstrate GGNN's ability to serve as a principled foundation for designing topology-aware GNNs.

## 1 INTRODUCTION

The concept of *neighborhood* plays a central role in most Graph Neural Network (GNN) architectures, serving as the foundation for message passing (Gilmer et al., 2017). This reliance is not arbitrary: neighborhoods provide comprehensive coverage of the graph structure, leveraging the adjacency matrix to facilitate efficient and systematic aggregation of local information. However, this local perspective comes with limitations. In particular, many GNNs have an expressive power bounded by the WL test (Sato, 2020), (Xu et al., 2019), limiting the ability of GNNs to capture broader topological structures.

To address the limitations of neighborhoods, researchers have proposed alternatives that incorporate richer structural information. One direction uses concepts from algebraic topology, such as simplicial complexes and higher-order faces, to capture interactions beyond pairs of nodes (Bodnar et al., 2021b), (Bodnar et al., 2021a), (Hajij et al., 2023), (Papillon et al., 2025). Another line of work relies on specific patterns or subgraphs (e.g., motifs) to encode characteristic structures as the basis for topologically-aware message passing (Bouritsas et al., 2023), (Ai et al., 2022). While these methods enrich the local perspective, they depend on handcrafted definitions or combinatorial choices. In contrast, neighborhoods themselves arise from a precise algebraic definition, suggesting that a more systematic algebraic generalization may provide a broader and more principled foundation.

We argue that an algebraic viewpoint provides such a foundation. Unlike topological constructs or handcrafted patterns, algebraic generalizations of neighborhoods can preserve their simplicity while extending their flexibility. Building on this insight, our work introduces an algebraic extension of neighborhoods that retains ease of use while enabling more expressive message-passing strategies. Our main contributions are as follows.

- **Algebraic generalization of neighborhoods**. We extend the conventional notion of neighborhoods by introducing the concept of *covers* for graphs. This generalization provides a principled and flexible foundation for understanding graph structure.

- **The GGNN framework**. We develop the *Grothendieck Graph Neural Networks* (GGNN) framework, an algebraic platform that creates, refines, and transforms covers into their matrix forms, recovering the adjacency matrix as a special case. GGNN offers a systematic way to design new message-passing strategies.

- **Sieve Neural Networks (SNN)**. As a concrete instantiation of GGNN, we introduce Sieve Neural Networks (SNN), based on a cover inspired by *sieves* in category theory. SNN exemplifies how an algebraic concept (sieves) can be translated into an architecture for message passing while preserving invariance.

- **Topology-aware evaluation (our probe)**. In addition to evaluating SNN on graph isomorphism task, we design a topology-encoding benchmark based on a special case of Label Propagation (LP) (Zhu & Ghahramani, 2002; Huang et al., 2020) with one propagation step ($\alpha = 1$). Here LP is used not as a learning algorithm but as a controlled probe to directly compare covers with neighborhoods. On citation networks and ogbn-arxiv, sieve covers consistently and substantially outperform the neighborhood cover, highlighting the advantage of covers in capturing topological structure in large graphs.

## 2 COVERS AND THEIR MATRIX INTERPRETATIONS

In this section we develop the notion of *covers* for graphs and show how to interpret them as matrices, laying the groundwork for the GGNN framework. We begin by assigning to each directed subgraph its matrix representation, establishing a bijection between directed subgraphs and their associated matrices. We then introduce two monoids: $\mathsf{Mod}(G)$, generated by directed subgraphs, and $\mathsf{Mom}(G)$, generated by their matrix representations. This allows us to extend the representation map to a monoidal homomorphism

$$\mathsf{Tr} : \mathsf{Mod}(G) \longrightarrow \mathsf{Mom}(G).$$

We prove that $\mathsf{Tr}$ is invariant under graph isomorphisms (via Change-of-Order mappings) and provides an algebraic description of a graph that is unique up to isomorphism. These results form the theoretical foundation of the GGNN framework.

### 2.1 MATRIX REPRESENTATIONS OF DIRECTED SUBGRAPHS

We consider undirected graphs $G = (V, E)$ whose node set $V$ is equipped with a fixed ordering. Our first step is to formalize *directed subgraphs* of $G$ and to define their matrix representations.

**Definition 2.1.1.** *(1) A **path** $p$ from node $v_{p_1}$ to node $v_{p_m}$ is an ordered sequence*

$$v_{p_1}, e_{p_1}, v_{p_2}, e_{p_2}, \ldots, v_{p_{m-1}}, e_{p_{m-1}}, v_{p_m},$$

*where each $e_{p_i}$ connects $v_{p_i}$ and $v_{p_{i+1}}$.*

*(2) A **directed subgraph** $D$ of $G$ is a connected, acyclic subgraph in which each edge is assigned a direction.*

**Neighborhoods as a special case.** A neighborhood can be seen as a directed subgraph obtained by orienting all incident edges *into* a fixed node (see Figure 4). In the adjacency matrix, each column corresponds to such a neighborhood; isolating the neighborhood of a node amounts to zeroing out the other columns.

**Matrix representation of a directed subgraph.** We extend the neighborhood-as-column view to arbitrary directed subgraphs by encoding *direction-respecting reachability*.

**Definition 2.1.2.** *Let $D$ be a directed subgraph of $G = (V, E)$.*

1. *Define a relation $\leq_D$ on $V$ by $v_i \leq_D v_j$ iff there exists a path in $D$ that respects edge directions and starts at $v_i$ and ends at $v_j$.*

2. *The matrix representation of $D$ is the $|V| \times |V|$ matrix $M_D$ with $(M_D)_{ij} = 1$ if $v_i \leq_D v_j$ and $(M_D)_{ij} = 0$ otherwise.*

The requirement that paths respect directions is essential: all walks counted by $M_D$ must follow the edge orientations in $D$. Intuitively, a directed subgraph specifies an *allowable message-flow pattern*; its matrix $M_D$ realizes this pattern. In this sense, matrices from directed subgraphs serve as structured alternatives to the standard adjacency matrix used in neighborhood-based message passing.

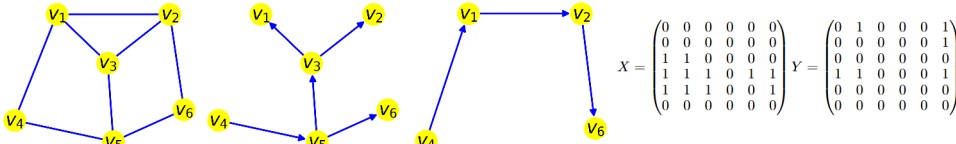

Figure 1: **Directed subgraphs and their matrices.** Two directed subgraphs of a graph $G$ (left) are shown: $\hat{D}$ (middle) and $\bar{D}$ (right). Their matrix representations are $X$ and $Y$, respectively. Each directed subgraph encodes a distinct strategy for propagating information; the matrices $X$ and $Y$ make these strategies directly usable in message passing.

**Representation map.** Definition 2.1.2 induces a map from directed subgraphs to matrices:

$$\mathsf{Rep} : \mathsf{DirSub}(G) \longrightarrow \mathsf{MatRep}(G),$$

where $\mathsf{DirSub}(G)$ is the set of directed subgraphs of $G$ and $\mathsf{MatRep}(G)$ is the image (subset) of $\mathsf{Mat}_{|V|}(\mathbb{R})$ consisting of matrices that arise from directed subgraphs via Definition 2.1.2.

**Theorem 2.1.3.** *The map $\mathsf{Rep}$ is an isomorphism between $\mathsf{DirSub}(G)$ and $\mathsf{MatRep}(G)$. In particular, each directed subgraph is uniquely determined by its matrix representation, and conversely every matrix in $\mathsf{MatRep}(G)$ corresponds to a unique directed subgraph.*

## 2.2 DEFINING COVERS FOR GRAPHS: AN ALGEBRAIC PLATFORM

While we can cover a graph $G$ by picking elements from $\mathsf{DirSub}(G)$ and map each to a matrix via $\mathsf{Rep}$, this space is limited: $\mathsf{DirSub}(G)$ is relatively small and its elements do not combine well. For instance, the union of the directed subgraphs $\hat{D}$ and $\bar{D}$ in Figure 1 is *not* a directed subgraph (multiple directed paths appear between some node pairs). Hence no matrix image exists for such a combination, which prevents its direct use in a message-passing scheme. This makes it hard to design diverse, meaningful strategies using only $\mathsf{DirSub}(G)$.

**Step 1: enlarge the space via a multigraph monoid.** To combine directed subgraphs systematically, we first endow them with an algebraic operation. A natural choice is to define $C \bigoplus D$ for $C, D \in \mathsf{DirSub}(G)$ as the *directed multigraph* obtained by taking the union of their node sets and the disjoint union of their directed edge sets. Let

$$\mathsf{Mult}(G) = \Big\{ \bigoplus_{i=1}^{k} D_i \; : \; k \geq 1, \; D_i \in \mathsf{DirSub}(G) \Big\}.$$

Then $(\mathsf{Mult}(G), \bigoplus)$ is a commutative monoid. This construction enlarges the object set, but it treats edges as independent and is largely *insensitive to path structure*: paths in $C \bigoplus D$ are simply concatenations of available directed edges, with little inheritance from the path sets of $C$ and $D$. As a result, $\bigoplus$ alone provides limited control for crafting new message-passing strategies.

**Step 2: add path sensitivity via a non-commutative monoid.** To encode which *paths* are allowed, not only which edges exist, we augment multigraphs with explicit path sets. Define

$$\mathsf{SMult}(G) = \big\{ (M, S) \; : \; M \in \mathsf{Mult}(G), \; S \subseteq \mathsf{Paths}(M) \big\},$$

and define a binary operation $\bullet$ by $(M, S) \bullet (N, T) = \big( M \bigoplus N, \; S \star T \big)$, where $S \star T$ is the union of the sets $S$, $T$, and the collection of paths constructed by the composition of paths in $S$ followed by paths in $T$. Since edges of $M \bigoplus N$ come from a disjoint union, paths in $S$ and $T$ remain disjoint subsets of $\mathsf{Paths}(M \bigoplus N)$.

**Theorem 2.2.1.** $(\mathsf{SMult}(G), \bullet)$ *is a non-commutative monoid.*

Non-commutativity comes from the order of path composition inside $\star$: if $(M, S)$ and $(N, T)$ have composable paths, then generally $(M, S) \bullet (N, T) \neq (N, T) \bullet (M, S)$; if they have no composable paths, $\star$ reduces to set union.

**Example 2.2.2.** *Let $d : u \to v$ and $e : v \to w$ be directed edges. Then $d \bullet e$ and $e \bullet d$ share the same edge multiset ($d \bigoplus e$), but differ in allowed paths: $d \bullet e$ contains a path from $u$ to $w$ (via $d$ then $e$), whereas $e \bullet d$ does not. Thus the order of composition matters.*

**Step 3: a representable submonoid for covers.** Elements $(M, S) \in \mathsf{SMult}(G)$ can be read as *message-passing strategies*: $M$ fixes the directed multigraph over which messages may travel, and $S$ specifies which directed paths are allowed. However, not every such pair admits a matrix representation compatible with an extension of Rep. To retain implementability, we restrict to the submonoid generated by genuine directed subgraphs, embedded via $D \mapsto \big(D, \mathsf{Paths}(D)\big)$.

**Definition 2.2.3.** *For a graph $G$, the* monoid of directed subgraphs *is the submonoid* $\mathsf{Mod}(G) \subseteq \mathsf{SMult}(G)$ *generated by* $\mathsf{DirSub}(G)$.

Hence each $(M, S) \in \mathsf{Mod}(G)$ can be written as

$$(M, S) = D_1 \bullet \cdots \bullet D_k, \qquad M = \bigoplus_{i=1}^{k} D_i, \qquad S = \mathsf{Paths}(D_1) \star \cdots \star \mathsf{Paths}(D_k).$$

In the next subsection we show that all elements of $\mathsf{Mod}(G)$ admit matrix representations via an extension of Rep, which makes them suitable for implementation.

**Definition 2.2.4.** *A* cover *of $G$ is a finite collection of elements of* $\mathsf{Mod}(G)$.

A cover thus specifies a family of message-passing strategies: each element constrains allowed paths locally, and the collection provides a flexible, task-dependent view of the graph (we do not require covering all nodes/edges). Because $\mathsf{Mod}(G)$ is infinite and $\bullet$ is non-commutative, this space is expressive enough to encode a wide range of perspectives on how information should flow. Moreover, the following result shows that directed edges suffice as generators, so complex strategies can be built compositionally.

**Theorem 2.2.5.** *Directed edges generate* $\mathsf{Mod}(G)$.

### 2.3 FROM COVERS TO MATRICES: AN ALGEBRAIC PERSPECTIVE

Our goal in this section is to extend the representation map Rep from directed subgraphs to arbitrary elements of $\mathsf{Mod}(G)$, so that a *cover* can be transformed into a *collection of matrices*. This requires a matrix-side operation that mirrors the path-composition on $\mathsf{Mod}(G)$.

**A monoid on matrices.** Let $\mathsf{Mat}_n(\mathbb{R})$ be the set of $n \times n$ real matrices. Define a binary operation

$$A \circ B := A + B + AB.$$

**Theorem 2.3.1.** $(\mathsf{Mat}_n(\mathbb{R}), \circ)$ *is a monoid.*

We now take the submonoid *generated* by $\mathsf{MatRep}(G)$ inside $(\mathsf{Mat}_{|V|}(\mathbb{R}), \circ)$.

**Definition 2.3.2.** *The* monoid of matrix representations *of $G$ is the submonoid* $(\mathsf{Mom}(G), \circ)$ *of* $(\mathsf{Mat}_{|V|}(\mathbb{R}), \circ)$ *generated by* $\mathsf{MatRep}(G)$.

**Extending** Rep **to covers.** Elements of $\mathsf{Mod}(G)$ are built by composing directed subgraphs with $\bullet$. The map below sends such compositions to matrices by replacing each directed subgraph $D_i$ with its matrix $\mathsf{Rep}(D_i)$ and each $\bullet$ with $\circ$.

**Theorem 2.3.3.** *The mapping* $\mathsf{Tr} : \mathsf{Mod}(G) \longrightarrow \mathsf{Mom}(G)$,

$$(M, S) = D_1 \bullet D_2 \bullet \cdots \bullet D_k \longmapsto A = A_1 \circ A_2 \circ \cdots \circ A_k,$$

*where $D_i \in \mathsf{DirSub}(G)$ and $A_i = \mathsf{Rep}(D_i)$, is a surjective monoidal homomorphism.*

**Interpretation (path counting).** In the proof of Theorem 2.3.3 one sees that $\mathsf{Tr}$ acts as a *path counter*: for $(M, S) \in \mathsf{Mod}(G)$, the $(i, j)$ entry of $\mathsf{Tr}(M, S)$ equals the number of paths in $S$ from $v_i$ to $v_j$. Thus, covers become *collections of matrices* in the same way that neighborhoods give rise to the adjacency matrix, but now with explicit control over composed paths. Although we were not able to show that $\mathsf{Tr}$ is an isomorphism, it extends an isomorphism on $\mathsf{DirSub}(G)$ and will be shown in the next subsection to characterize graphs up to isomorphism, supporting its use as a faithful matrix transformation of covers.

**Example 2.3.4.** *In Figure 1, let $\hat{D}$ and $\bar{D}$ be two directed subgraphs of $G$ with matrix representations $X$ and $Y$. Using $\bullet$, we may form new strategies $\hat{D} \bullet \bar{D}$ and $\bar{D} \bullet \hat{D}$. Their matrix transforms are obtained as follows:*

$$\mathsf{Tr}(\hat{D} \bullet \bar{D}) \;=\; \mathsf{Tr}(\hat{D}) \circ \mathsf{Tr}(\bar{D}) \;=\; X \circ Y, \qquad \mathsf{Tr}(\bar{D} \bullet \hat{D}) \;=\; \mathsf{Tr}(\bar{D}) \circ \mathsf{Tr}(\hat{D}) \;=\; Y \circ X.$$

*Since $\circ$ is generally non-commutative in this context, the two results differ, reflecting the order-sensitivity of path composition in the underlying cover construction.*

## 3 GROTHENDIECK GRAPH NEURAL NETWORKS FRAMEWORK

### 3.1 ALGEBRAIC FOUNDATIONS OF GRAPHS

We have introduced two monoids associated with a graph and a monoidal homomorphism between them. We now ask: *to what extent do these algebraic objects describe the underlying graph?* To address this, we first formalize how reordering node indices acts on the matrix space and verify that this action is compatible with our monoidal structures.

**Change-of-Order mappings.** A matrix $A \in \mathsf{Mat}_n(\mathbb{R})$ represents a linear map $\mathbb{R}^n \to \mathbb{R}^n$ with respect to the standard basis. If we reorder the standard basis of $\mathbb{R}^n$ (equivalently, relabel the coordinates), the matrix representation of the same linear map is obtained by reindexing the rows and columns of $A$. Any linear isomorphism $f : \mathsf{Mat}_n(\mathbb{R}) \to \mathsf{Mat}_n(\mathbb{R})$ arising in this way is called a **Change-of-Order mapping** (see Example C.0.3). Intuitively, this captures the effect of node relabeling at the matrix level.

**Proposition 3.1.1.** *Suppose $f : \mathsf{Mat}_n(\mathbb{R}) \to \mathsf{Mat}_n(\mathbb{R})$ is a Change-of-Order mapping. Then $f$ preserves the standard algebraic operations on matrices: it is compatible with monoidal operation $\circ$, with matrix multiplication, and with element-wise (Hadamard) multiplication.*

**Relating graph isomorphisms and algebra.** A graph isomorphism $f : G \to H$ is a bijection on nodes that preserves edges, and therefore corresponds to a reordering of node indices. Hence it induces a Change-of-Order mapping $\mathsf{CO}(f) : \mathsf{Mat}_{|V_G|}(\mathbb{R}) \to \mathsf{Mat}_{|V_H|}(\mathbb{R})$. The next result shows that this relabeling is compatible with our monoidal constructions and with the translation to matrices.

**Theorem 3.1.2.** *Every graph isomorphism $f : G \to H$ induces monoidal isomorphisms $\mathsf{Mod}(f) : \mathsf{Mod}(G) \to \mathsf{Mod}(H)$ and $\mathsf{Mom}(f) : \mathsf{Mom}(G) \to \mathsf{Mom}(H)$ such that the following diagram commutes, where $\iota$ denotes the inclusions:*

$$
\begin{array}{ccccc}
\mathsf{Mod}(G) & \xrightarrow{\;\mathsf{Tr}_G\;} & \mathsf{Mom}(G) & \overset{\iota}{\hookrightarrow} & \mathsf{Mat}_{|V_G|}(\mathbb{R}) \\
{\scriptstyle \mathsf{Mod}(f)}\big\downarrow & & {\scriptstyle \mathsf{Mom}(f)}\big\downarrow & & \big\downarrow{\scriptstyle \mathsf{CO}(f)} \\
\mathsf{Mod}(H) & \xrightarrow[\;\mathsf{Tr}_H\;]{} & \mathsf{Mom}(H) & \underset{\iota}{\hookrightarrow} & \mathsf{Mat}_{|V_H|}(\mathbb{R})
\end{array}
\tag{1}
$$

**A converse direction.** We also have a partial converse: if a Change-of-Order mapping identifies the matrix-level monoids of two graphs, then the graphs are isomorphic.

**Theorem 3.1.3.** *Let $G$ and $H$ be graphs with $|V_G| = |V_H| = n$, and let $f : \mathsf{Mat}_n(\mathbb{R}) \to \mathsf{Mat}_n(\mathbb{R})$ be a Change-of-Order mapping. If the restriction of $f$ to $\mathsf{Mom}(G)$ is an isomorphism onto $\mathsf{Mom}(H)$, then $G$ and $H$ are isomorphic.*

### 3.2 DEFINITION OF THE GGNN FRAMEWORK

Theorems 3.1.2 and 3.1.3 establish the key invariance principle underlying our construction. In Diagram 1, an isomorphism between graphs $G$ and $H$ corresponds to the vertical homomorphisms

being isomorphisms; equivalently, a relabeling of nodes in a graph induces isomorphic transformations both on a cover and on its matrix transformation. Consequently, the *horizontal* arrows in the diagram provide an algebraic description that is unique up to graph isomorphism. Leveraging this observation, we formalize the *Grothendieck Graph Neural Networks (GGNN)* framework as the following algebraic pipeline:

**Definition 3.2.1.** *For a graph $G = (V, E)$, the GGNN framework is the composition*

$$\mathsf{Mod}(G) \xrightarrow{\ \mathsf{Tr}\ } \mathsf{Mom}(G) \overset{\iota}{\hookrightarrow} \mathsf{Mat}_{|V|}(\mathbb{R}) \tag{2}$$

**How GGNN is used.**   The framework exposes three actions: (i) *choose* a cover in $\mathsf{Mod}(G)$; (ii) *translate* it to matrices via $\mathsf{Tr}$; (iii) optionally *enrich* the resulting collection inside $\mathsf{Mat}_{|V|}(\mathbb{R})$ using the allowed operations from Proposition 3.1.1. See Appendix D for more details.

**Neighborhoods as a special case.**   The framework recovers standard message passing from neighborhoods:

**Theorem 3.2.2.** *The collection of neighborhoods forms a cover in $\mathsf{Mod}(G)$ and, under $\mathsf{Tr}$, maps to the adjacency matrix in $\mathsf{Mat}_{|V|}(\mathbb{R})$.*

**Remarks.**   (i) A graph can be characterized by the submonoid generated by its neighborhoods; see Appendix G.1. (ii) We compare GGNN with higher-order GNN families in Appendix E.

## 4   SIEVE NEURAL NETWORKS: A MODEL WITHIN THE GGNN FRAMEWORK

Inspired by *sieves* in category theory (MacLane & Moerdijk, 1994), we instantiate the GGNN framework with a concrete cover that yields the *Sieve Neural Network* (SNN). For each node $v$, we construct a family of outward-expanding substructures, formalized as elements of $\mathsf{Mod}(G)$, which together form a *sieve-inspired cover* of the graph. This cover generalizes the standard neighborhood view by admitting multiple direction-respecting pathways for information exchange. Via the matrix map $\mathsf{Tr}$, the cover is translated into operators used for message passing, exposing richer topological relationships than adjacency-based aggregation while remaining permutation-consistent. In what follows, we define the sieve and cosieve elements, assemble the cover, and derive the SNN architecture from their matrix transformations.

### 4.1   COVER OF SIEVES FOR GRAPHS

**Constructing sieve elements in** $\mathsf{Mod}(G)$**.**   Fix a node $v$ and build breadth-first "layers" around $v$:

$$N_0(v) = \{v\}, \qquad N_1(v) = N(v), \qquad N_k(v) = \Big( \bigcup_{u \in N_{k-1}(v)} N(u) \Big) \setminus \bigcup_{i=0}^{k-1} N_i(v) \quad (k \geq 2).$$

For each $k \geq 1$, orient all edges *toward* $v$ across consecutive layers and collect them as

$$M_k(v) = \{\, w \to u \ : \ wu \in E, \ w \in N_k(v), \ u \in N_{k-1}(v) \,\}, \qquad M_0(v) = \varnothing.$$

Edges in the same $M_k(v)$ are pairwise non-composable (each goes from layer $k$ to $k-1$), so the order in which they are combined is irrelevant. Define

$$D_k(v) \ := \ \underset{e \in M_k(v)}{\bullet}\, e$$

and assemble the depth-$k$ *sieve element*

$$\mathsf{Sieve}(v, k) \ := \ D_k(v) \bullet D_{k-1}(v) \bullet \cdots \bullet D_1(v) \bullet D_0(v),$$

where $D_0(v)$ is the identity of $\mathsf{Mod}(G)$; see Figure 2. Since the layers eventually empty, there exists $k_0$ with $N_{k_0+1}(v) = \varnothing$, hence $\mathsf{Sieve}(v, k)$ stabilizes for $k \geq k_0$. We denote this saturated element by $\mathsf{Sieve}(v, -1) := \mathsf{Sieve}(v, k_0)$. To construct the *opposite* sieve, reverse the directions in each $M_k(v)$: let $M_k^{\mathrm{op}}(v)$ be $M_k(v)$ with all edges reversed and set

$$D_k^{\mathrm{op}}(v) \ := \ \underset{e \in M_k^{\mathrm{op}}(v)}{\bullet}\, e, \qquad \mathsf{CoSieve}(v, \ell) \ := \ D_0^{\mathrm{op}}(v) \bullet D_1^{\mathrm{op}}(v) \bullet \cdots \bullet D_\ell^{\mathrm{op}}(v).$$

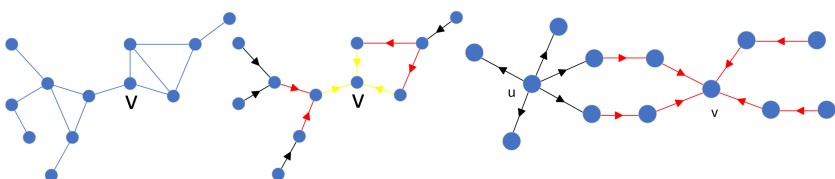

Figure 2: **Sieve construction.** Left: a graph $G$. Middle: $\mathsf{Sieve}(v, 3) = D_3(v) \bullet D_2(v) \bullet D_1(v)$ around $v$; yellow, red, and black edges indicate $D_1(v)$, $D_2(v)$, and $D_3(v)$, respectively. Right: a graph $H$; the element $\mathsf{CoSieve}(u, 1) \bullet \mathsf{Sieve}(v, 2) \in \mathsf{Mod}(H)$ specifies allowed interactions between $u$ and $v$ in $\mathsf{SNN}(\alpha, (1, 2))$.

**The cover of sieves.** For every node $v$ in $G$, collect all depth-truncated and saturated sieve and co-sieve elements:

$$\mathsf{Sieve}(v, 0), \; \mathsf{Sieve}(v, 1), \; \ldots, \; \mathsf{Sieve}(v, -1) \quad \text{and} \quad \mathsf{CoSieve}(v, 0), \; \mathsf{CoSieve}(v, 1), \; \ldots, \; \mathsf{CoSieve}(v, -1).$$

The *cover of sieves* is the finite collection containing these elements for all nodes $v$ in $G$.

**Matrix interpretation of the cover of sieves.** Apply the monoidal homomorphism $\mathsf{Tr}$ to obtain matrices:

$$\mathsf{Image}(v, k) := \mathsf{Tr}\big(\mathsf{Sieve}(v, k)\big), \qquad \mathsf{CoImage}(v, \ell) := \mathsf{Tr}\big(\mathsf{CoSieve}(v, \ell)\big).$$

Since $\mathsf{Tr}$ is monoidal,

$$
\begin{aligned}
\mathsf{Image}(v, k) &= \mathsf{Tr}\big(D_k(v) \bullet D_{k-1}(v) \bullet \cdots \bullet D_0(v)\big) \\
&= \mathsf{Tr}\big(D_k(v)\big) \circ \mathsf{Tr}\big(D_{k-1}(v)\big) \circ \cdots \circ \mathsf{Tr}\big(D_0(v)\big).
\end{aligned}
\tag{3}
$$

Within a fixed layer $i$, edges in $M_i(v)$ are not composable, so for distinct $e, c \in M_i(v)$ we have $\mathsf{Tr}(e)\,\mathsf{Tr}(c) = \mathsf{Tr}(c)\,\mathsf{Tr}(e) = 0$. Hence, by Theorem H.5.1,

$$\mathsf{Tr}\big(D_i(v)\big) = \mathsf{Tr}\Big(\underset{e \in M_i(v)}{\bullet}\, e\Big) = \underset{e \in M_i(v)}{\circ}\, \mathsf{Tr}(e) = \sum_{e \in M_i(v)} \mathsf{Tr}(e),$$

i.e., $\mathsf{Tr}\big(D_i(v)\big)$ is obtained from the adjacency matrix by keeping only entries corresponding to directed edges in $M_i(v)$. Moreover, $\mathsf{CoImage}(v, \ell) = \mathsf{Image}(v, \ell)^\top$ (transpose), so it suffices to compute one of them. Algorithms for these computations are given in Appendix F.4. In addition, Appendix G.2 shows that the submonoid generated by this cover determines the graph.

**Invariance.** The cover of sieves is stable under graph isomorphisms, and the induced matrices transform accordingly.

**Theorem 4.1.1.** *If $f : G \to H$ is a graph isomorphism, then $\mathsf{Mod}(f)\big(\mathsf{Sieve}(v, k)\big) = \mathsf{Sieve}\big(f(v), k\big)$ and $\mathsf{Mom}(f)\big(\mathsf{Image}(v, k)\big) = \mathsf{Image}\big(f(v), k\big)$.*

### 4.2 DESIGN AND CONSTRUCTION OF THE MODEL

Building on the cover of sieves and its matrix interpretation, we define the *Sieve Neural Network* (SNN) in two variants of increasing flexibility. A detailed comparison with MPNNs is provided in Appendix F.

**Variant** $\mathsf{SNN}(\alpha, (l, k))$**.** In the $\alpha$ variant, for each ordered pair of nodes $(v_i, v_j)$ we interpret $\mathsf{CoSieve}(v_i, l)$ as a *sender* and $\mathsf{Sieve}(v_j, k)$ as a *receiver*. The admissible transmissions from $v_i$ to $v_j$ are precisely the paths allowed by the composed element $\mathsf{CoSieve}(v_i, l) \bullet \mathsf{Sieve}(v_j, k)$, (see Figure 2). Under the map $\mathsf{Tr}$, the number of such paths equals the $(i, j)$ entry of $\mathsf{CoImage}(v_i, l) \circ \mathsf{Image}(v_j, k)$. To obtain a scale-aware score, we normalize by the *sending capacity* of $v_i$ and the *receiving capacity* of $v_j$: let

$$r_i = \sum_q \big(\mathsf{CoImage}(v_i, l)\big)_{iq}, \qquad c_j = \sum_p \big(\mathsf{Image}(v_j, k)\big)_{pj}.$$

Then the output matrix of $\mathsf{SNN}(\alpha, (l, k))$ on $G$ is $S^{(\alpha)}$ with $S_{ij}^{(\alpha)} = \dfrac{\left(\mathsf{CoImage}(v_i, l) \circ \mathsf{Image}(v_j, k)\right)_{ij}}{r_i\, c_j}$.

Intuitively, $S_{ij}^{(\alpha)}$ is the fraction of realized paths relative to a node-pair–specific capacity. If the normalization is omitted, we denote the model by $\mathsf{SNN}_o(\alpha, (l, k))$.

**Variant** $\mathsf{SNN}(\beta, (l_1, \ldots, l_t))$. The $\beta$ variant aggregates information globally by summing over node-centered images and co-images, alternating sender/receiver roles across depths. The model output is $S^{(\beta)} = Su_1 \circ Su_2 \circ \cdots \circ Su_t$, where $Su_i = \sum_{v \in V} \mathsf{CoImage}(v, l_i)$ for odd $i$ and $Su_i = \sum_{v \in V} \mathsf{Image}(v, l_i)$ for even $i$, with $1 \le i \le t$.

**How** $\mathsf{SNN}$ **is used.** $\mathsf{SNN}$ is applied *once* as a preprocessing step to transform each graph: we replace its adjacency matrix with the corresponding $\mathsf{SNN}$ output (either $S^{(\alpha)}$ or $S^{(\beta)}$). The transformed dataset can then be fed to any message-passing GNN, substituting the traditional neighborhood cover with the sieve cover. The *time complexity* of this transformation is analyzed in Appendix F.3.

**Invariance.** The model respects graph isomorphisms (node relabelings), making it suitable for graph isomorphism and classification tasks.

**Theorem 4.2.1.** $\mathsf{SNN}$ *is invariant under graph isomorphism.*

## 5 EXPERIMENTS

To show how a shift in perspective improves graph understanding, we conduct a comprehensive evaluation of $\mathsf{SNN}$ on two tasks: (i) graph isomorphism and (ii) a topology-encoding probe.

### 5.1 GRAPH ISOMORPHISM.

We evaluate $\mathsf{SNN}$ on standard isomorphism benchmarks to test its ability to distinguish non-isomorphic graphs and to compare against WL-style limits. Throughout, we use the $\beta$ variants in strong (saturated) settings, and we do *not* train any parameters: each graph $G$ is mapped once to an $\mathsf{SNN}$ output matrix $S(G)$, from which we compute simple permutation-invariant summaries as embeddings. By invariance, isomorphic graphs yield identical embeddings; non-identical embeddings imply non-isomorphism.[1]

**SR (Strongly Regular graphs).** We use **all** publicly available collections of strongly regular graphs from Brendan McKay's Graph Data (archived). SR graphs are challenging since 3-WL cannot fully distinguish them (Bodnar et al., 2021b). For each collection, we apply $\mathsf{SNN}(\beta, (-1, -1, -1))$ to every graph $G$ and compute a 6-dimensional embedding from $S(G)$: $\big(\det(S),\ \mathrm{Min}(S),\ \mathrm{Mean}(S),\ \mathrm{Var}(S),\ \mathrm{Mean}(\mathrm{diag}(S)),\ \mathrm{Var}(\mathrm{diag}(S))\big)$. By invariance and Theorem 3.1.2, $\det(S)$ is preserved under relabeling, so isomorphic graphs match. Within each collection, $\mathsf{SNN}$ assigns distinct embeddings to *all* graphs, yielding a $0\%$ failure rate; see Table 1.

**CSL (Circular Skip Links).** CSL contains 150 4-regular graphs partitioned into 10 isomorphism classes and is widely used to probe GNN expressivity (Murphy et al., 2019; Dwivedi et al., 2023). We run $\mathsf{SNN}(\beta, (-1))$ on each graph and use $\mathrm{Sum}(S)$ (sum of all entries of $S$) as a permutation-invariant scalar embedding. The resulting values perfectly separate the 10 classes: graphs within a class share the same value; graphs from different classes do not.

**BREC.** BREC (Wang & Zhang, 2024) contains 400 pairs of non-isomorphic graphs divided into four categories (60 Basic, 140 Regular, 100 Extension, and 100 CFI), with cases that remain indistinguishable even under the 4-WL test. For each graph $G$, we apply $\mathsf{SNN}(\beta, (-1, -1, -1, -1))$ and construct the same type of embedding as used in the **SR** experiment. Across all BREC pairs, $\mathsf{SNN}$ consistently assigns distinct embeddings to the two graphs in each pair, yielding a $0\%$ failure rate (Table 1).

---

[1]In practice we compare embeddings with a small numerical tolerance. The depth "$-1$" denotes the saturated sieve (Section 4).

Table 1: Left: Failure rates of 3-WL and SNN across Strongly Regular graphs. Right: Number of distinguished pairs on **BREC**. Baseline values from Wang & Zhang (2024).

| Graph Category | 3-WL (%) | SNN (%) | Graph Category | 3-WL (%) | SNN (%) | Model | Basic (60) | Reg. (140) | Ext. (100) | CFI (100) |
|---|---|---|---|---|---|---|---|---|---|---|
| SRG(25,12,5,6) | 100 | 0 | SRG(36,15,6,6) | 100 | 0 | 3-WL | 60 | 50 | 100 | 60 |
| SRG(26,10,3,4) | 100 | 0 | SRG(37,18,8,9) | 100 | 0 | SSWL-P | 60 | 50 | 100 | 38 |
| SRG(28,12,6,4) | 100 | 0 | SRG(40,12,2,4) | 100 | 0 | I²-GNN | 60 | 100 | 100 | 21 |
| SRG(29,14,6,7) | 100 | 0 | SRG(65,32,15,16) | 100 | 0 | GSN | 60 | 99 | 95 | 0 |
| SRG(35,16,6,8) | 100 | 0 | | | | PPGN | 60 | 50 | 100 | 23 |
| SRG(35,18,9,9) | 100 | 0 | | | | SNN | 60 | 140 | 100 | 100 |
| SRG(36,14,4,6) | 100 | 0 | | | | | | | | |

## 5.2 Topology Encoding (probe).

Our aim here is to evaluate *only* the topology encoded by SNN as a preprocessing operator, independently of any learnable parameters or downstream training. To do so, we design a parameter-free *probe* based on one-step Label Propagation (LP) (Zhu & Ghahramani, 2002; Huang et al., 2020) with $\alpha = 1$. This choice isolates the structural signal present in the propagation operator and avoids confounds from optimization, regularization, or model capacity. We compare (i) the *neighborhood cover* (the adjacency matrix) against (ii) *cover of sieves* induced by SNN. Concretely, we run LP on $A \in \{$ Adj, $\mathsf{SNN}(\beta, (1)), \mathsf{SNN}(\beta, (1,1)), \mathsf{SNN}(\beta, (1,1,1)) \}$. By construction, $\mathsf{SNN}(\beta, (1))$ coincides with the adjacency matrix and serves as the neighborhood baseline; the deeper sequences $(1,1)$ and $(1,1,1)$ produce sieve-based operators that go beyond neighborhoods.

**LP update (one step, no learning).** Given initial one-hot labels $Y^{(0)}$ on the training nodes (zeros elsewhere), we propagate once: $Y^{(1)} = \widehat{M} Y^{(0)}$. Here $\widehat{M}$ is a normalized version of the chosen operator $A$. We report results for three standard normalizations $\widehat{M} \in \{ DAD, DA, AD \}$, where $D$ is the degree matrix induced by $A$. This ensures comparability across covers and conforms to standard LP practice. We evaluate on **Cora**, **CiteSeer**, **PubMed**, and **ogbn-arxiv**; the dataset specifications and the runtimes of models $\mathsf{SNN}(\beta, (1,1))$ and $\mathsf{SNN}(\beta, (1,1,1))$ are reported in Table 3. Because LP has no learnable parameters, any performance difference directly reflects the topology encoded by the operator $A$.

Table 2: Test accuracy of Label Propagation (1 step, $\alpha = 1$)

| | **Adjacency** | | | $\mathsf{SNN}(\beta, (1,1))$ | | | $\mathsf{SNN}(\beta, (1,1,1))$ | | |
|---|---|---|---|---|---|---|---|---|---|
| **Dataset** | DAD | DA | AD | DAD | DA | AD | DAD | DA | AD |
| Cora | 0.2600 | 0.2600 | 0.2580 | 0.5120 | 0.5050 | 0.5070 | 0.6090 | 0.6020 | 0.6080 |
| CiteSeer | 0.1370 | 0.1370 | 0.1370 | 0.2610 | 0.2580 | 0.2590 | 0.3700 | 0.3680 | 0.3720 |
| PubMed | 0.1890 | 0.1890 | 0.1890 | 0.2480 | 0.2480 | 0.2460 | 0.4230 | 0.4290 | 0.4240 |
| ogbn-arxiv | 0.6173 | 0.5969 | 0.6125 | 0.6627 | 0.6449 | 0.6252 | 0.6469 | 0.6416 | 0.5811 |

Across all datasets, one-step LP achieves its lowest accuracy with the neighborhood cover (adjacency), and substantially higher accuracy with sieve covers. On **Cora**, accuracy rises from $\approx 26\%$ (adjacency) to $> 50\%$ with $\mathsf{SNN}(\beta, (1,1))$ and to $> 60\%$ with $\mathsf{SNN}(\beta, (1,1,1))$; on **CiteSeer** and **PubMed**, sieve covers more than double the adjacency baseline; and on **ogbn-arxiv**, sieve covers also yield notable gains. Since the probe has *no learnable parameters*, these improvements can only come from richer topology captured by sieve-based operators. This provides direct evidence that SNN, used purely as preprocessing, encodes topological relationships beyond those available from neighborhood covers.

## 6 Conclusion

We formalized *covers* as an algebraic generalization of neighborhoods and introduced the GGNN framework to systematically design covers and translate them into matrices, recovering the adjacency matrix as a special case. This platform simplifies model construction. As a concrete instance, we proposed Sieve Neural Networks (SNN), which operationalize the framework and show strong performance on graph isomorphism and topology-encoding probes. Looking ahead, we will deepen the analysis of GGNN's expressive power and applications, including a more comprehensive theoretical comparison between SNN and the Weisfeiler–Lehman test.

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

## A   THE USE OF LARGE LANGUAGE MODELS (LLMS)

We employed large language models (LLMs) to refine the writing and improve the grammar of this paper, with the goal of enhancing clarity and readability. All ideas, scientific content, definitions, theorems, proofs, and experimental results were conceived and developed entirely by the authors.

## B   RELATED WORK

Many classical GNN architectures can be unified under the neighborhood-based *Message Passing Neural Network* (MPNN) paradigm Gilmer et al. (2017). A large body of work seeks to move beyond strict 1-hop neighborhoods by altering the graph on which messages are passed or by enriching the operators/features used for aggregation.

**Passing messages on derived graphs.** One line of work replaces the original graph with a derived graph and then applies MPNN. For example, Gasteiger et al. (2021) constructs the *directed line graph*, whose nodes correspond to directed edges of the original graph and where two nodes are adjacent if the underlying edges share an endpoint; message passing is then performed on this derived graph. In Ai et al. (2022), each graph is mapped to a *topology-level* summary graph built from subgraphs; message passing runs jointly on the original graph and its summary, making the propagation explicitly topology-aware.

**Substructure- and kernel-based encodings.** Another direction injects information about motifs or subgraphs. Graph Substructure Networks (GSNs) Bouritsas et al. (2023) enrich node/edge features with positions within selected patterns, integrating substructure signals into message passing. KerGNNs Feng et al. (2022a) use small graphs as filters—via graph kernels such as random-walk kernels—applied to node-centered subgraphs; replacing the raw neighborhood with a filtered subgraph can increase expressivity over vanilla MPNNs.

**Contextual and multi-hop neighborhoods.** Contextualization beyond the immediate neighborhood is also common. ID-GNN You et al. (2021) attends to occurrences of a node within its ego network, effectively differentiating its roles across contexts. Extensions to $k$-hop neighborhoods Feng et al. (2022b) aggregate information from larger receptive fields; KP-GNN further selects $k$-hop neighbors via shortest-path or random-walk kernels, yielding a framework that can surpass standard MPNNs.

**Local topology operators and stochastic perturbations.** In Vignac et al. (2020), each node is associated with a *local context* matrix intended to capture surrounding topology; these contexts replace raw features during message passing and have been shown effective on topology-sensitive tasks (e.g., cycle detection), outperforming MPNNs in those settings. The approach in Papp et al. (2021) applies message passing to randomly thinned graphs obtained by deleting each node with small probability and aggregates the outcomes, preserving much of the original topology while introducing beneficial stochasticity.

**Topological deep learning.** Tools from algebraic topology provide higher-order generalizations of graphs that encode multi-level interactions. Works based on simplicial and CW complexes Bodnar et al. (2021b;a) replace node–edge neighborhoods with higher-dimensional cells and associated incidence structures, yielding message-passing schemes that explicitly reason over topology beyond pairwise relations.

## C    DEFINITIONS AND EXAMPLES

The definition of a monoid and monoidal homomorphism are as follows (Hungerford, 1980):

**Definition C.0.1.** *A monoid is a non-empty set* M *together with a binary operation* $\cdot$ *on* M *which*

    *1) is associative:* $a \cdot (b \cdot c) = (a \cdot b) \cdot c$ *for all* $a, b, c \in$ M *and*

    *2) contains identity element* $e \in$ M *such that* $a \cdot e = e \cdot a = a$

*If, for all* $a, b \in$ M, *the operation satisfies* $a \cdot b = b \cdot a$, *then we say that* M *is a commutative monoid.*

**Definition C.0.2.** *A* monoid homomorphism *between monoids* $(M, \bullet)$ *and* $(N, \circ)$ *with identity elements* $e_M$ *and* $e_N$, *respectively, is a function* $f : M \to N$ *such that*

$$f(x \bullet y) \;=\; f(x) \circ f(y) \quad \text{for all } x, y \in M, \qquad f(e_M) = e_N.$$

**Example C.0.3.** *Considering a Change-of-Order mapping* $f : \mathsf{Mat}_3(\mathbb{R}) \to \mathsf{Mat}_3(\mathbb{R})$, *obtained by reordering the standard basis* $\{e_1, e_2, e_3\}$ *to the basis* $\{e_3, e_2, e_1\}$. *For a given matrix* $A$, *we get the matrix* $f(A)$ *as follows:*

$$A \longmapsto f(A)$$

$$
\begin{array}{c}
\begin{array}{ccc} e_1 & e_2 & e_3 \end{array} \\
\begin{array}{c} e_1 \\ e_2 \\ e_3 \end{array}
\begin{pmatrix} a_{11} & a_{12} & a_{13} \\ a_{21} & a_{22} & a_{23} \\ a_{31} & a_{32} & a_{33} \end{pmatrix}
\end{array}
\xmapsto{f : e_1 \leftrightarrow e_3}
\begin{array}{c}
\begin{array}{ccc} e_3 & e_2 & e_1 \end{array} \\
\begin{array}{c} e_3 \\ e_2 \\ e_1 \end{array}
\begin{pmatrix} a_{33} & a_{32} & a_{31} \\ a_{23} & a_{22} & a_{21} \\ a_{13} & a_{12} & a_{11} \end{pmatrix}
\end{array}
$$

# D    EXPLANATION FOR CONSTRUCTING A MODEL IN GGNN FRAMEWORK

The process of designing a GNN model within this framework is outlined as follows:

1) For a given graph $G$, the process involves selecting a collection $\mathcal{C}_G$ of elements from $\mathsf{Mod}(G)$ to serve as a cover for $G$. These elements can be generated using $\mathsf{DirSub}(G)$ and the binary operation $\bullet$. Notably, Theorem 2.2.5 ensures the ability to create any suitable and desired elements by leveraging directed edges and the operator $\bullet$.

2) Next, the chosen cover is transformed into a collection of matrices within $\mathsf{Mom}(G)$, utilizing $\mathsf{Tr}$. During this transformation, the operation $\circ$ and other elements of $\mathsf{Mom}(G)$ can be employed to convert the original collection into a new one. The resulting output at this stage is denoted by $\mathcal{A}_G$.

3) By utilizing $\iota$, the collection obtained in the second stage transitions into a larger and more equipped space, a suitable environment for enrichment. This stage leverages all the operations outlined in Proposition 3.1.1 to complete the model's design. Following the processing of $\mathcal{A}_G$ in this stage, we obtain a new collection of matrices denoted by $\mathcal{M}_G$, representing the model's output.

Hence, a model is a mapping that associates a collection of matrices $\mathcal{M}_G$ with a given graph $G$. $\mathcal{M}_G$ plays a role akin to the adjacency matrix and provides an interpretation of the chosen cover for use in various forms of message passing. While the second and third stages can be merged, we prefer to emphasize the significance of $\mathsf{Tr}$ in this process.

This construction of a model is appropriate for tasks such as node classification. For graph classification, we need an invariant construction. Based on Theorem 3.1.2, a graph isomorphism $f : G \rightarrow H$ transform the triple $(\mathcal{C}_G, \mathcal{A}_G, \mathcal{M}_G)$ to a triple $(\mathcal{C}'_H, \mathcal{A}'_H, \mathcal{M}'_H)$ for graph $H$ and this may be different from $(\mathcal{C}_H, \mathcal{A}_H, \mathcal{M}_H)$. So a model constructed in the GGNN framework is invariant if for every graph isomorphism $f : G \rightarrow H$, the maps $\mathsf{Mod}(f)$, $\mathsf{Mom}(f)$ and $\mathsf{CO}(f)$ induce one-to-one correspondences between $\mathcal{C}_G$ and $\mathcal{C}_H$, $\mathcal{A}_G$ and $\mathcal{A}_H$, and $\mathcal{M}_G$ and $\mathcal{M}_H$, respectively. The model SNN is an example of an invariant model.

# E    GGNN FRAMEWORK VS. HIGHER-ORDER GNNS: A COMPARISON

We contrast GGNN with higher-order GNNs such as MPSN Bodnar et al. (2021b), CWN Bodnar et al. (2021a), GSN Bouritsas et al. (2023), and TLGNN Ai et al. (2022).

**Framework, not a single model.**    GGNN is a *design framework*: it gives precise, graph-agnostic definitions of *covers* (generalizing neighborhoods) and a principled way to turn them into matrices. Whereas higher-order GNNs typically hard-code one specific alternative to neighborhood aggregation, GGNN provides an *infinite design space* of covers, of which the standard neighborhood cover is a special case, enabling diverse message-passing strategies tailored to a task.

**Topology-aware by construction.**    By Theorems 3.1.2 and 3.1.3, GGNN yields an algebraic description of a graph that is unique up to isomorphism. Each monoidal element of $\mathsf{Mod}(G)$ encodes concrete topological relationships; choosing a cover selects which aspects of topology to expose to downstream GNNs. Moreover, the algebra (composition, translation to matrices) lets one combine ideas from other paradigms within a single coherent toolkit.

**Example: recovering $k$-hop message passing.**    GGNN can reproduce common higher-order behaviors. Starting from the neighborhood cover $\{S_v : v \in G\}$, define for a node $v_k$ the set

$$\text{2-hop}(v_k) = \big\{\, S_{v_{k_i}} \;\bullet\; e_i \;:\; v_{k_i} \in N(v_k),\; e_i : v_{k_i} \rightarrow v_k \,\big\}.$$

Let $\text{2-hop}(G) = \bigcup_{v_k} \text{2-hop}(v_k)$. Applying $\mathsf{Tr}$ maps this cover to a collection of matrices, which can be aggregated (e.g., by summation) to obtain a 2-hop propagation operator, mirroring the effect of $k$-hop message passing in Feng et al. (2022b).

# F FURTHER DETAILS ON SNN

## F.1 MODEL EXPLANATIONS

The SNN construction provides two ways to collapse the matrix collection induced by the sieve cover into a single operator: the $\alpha$- and $\beta$-variants.

**$\alpha$-variant.** Using $\mathsf{ColImage}(v, l) = \mathsf{Image}(v, l)^\top$, we obtain

$$\mathsf{ColImage}(v_i, l) \circ \mathsf{Image}(v_j, k) = \big( \mathsf{ColImage}(v_j, k) \circ \mathsf{Image}(v_i, l) \big)^\top.$$

Hence the output of $\mathsf{SNN}(\alpha, (l, k))$ is the transpose of the output of $\mathsf{SNN}(\alpha, (k, l))$, and $\mathsf{SNN}(\alpha, (l, l))$ is symmetric. For $l \neq k$, symmetry need not hold (cf. Example F.2.1), so $\mathsf{SNN}(\alpha, (l, k))$ and $\mathsf{SNN}(\alpha, (k, l))$ may differ. Moreover, increasing the radii only adds admissible paths: if $l \leq l'$ and $k \leq k'$, then $\mathsf{SNN}(\alpha, (l', k'))$ captures (entrywise) at least as many paths as $\mathsf{SNN}(\alpha, (l, k))$.

**$\beta$-variant.** The families $\{\mathsf{Sieve}(v, l_i)\}_v$ (or $\{\mathsf{CoSieve}(v, l_i)\}_v$) form subcovers of the cover of sieves. Their matrix summaries

$$Su_i = \sum_{v \in V} \mathsf{Image}(v, l_i) \quad \text{or} \quad Su_i = \sum_{v \in V} \mathsf{ColImage}(v, l_i)$$

aggregate all allowed paths contributed by the chosen subcover. Composing these summaries with the monoid operation $\circ$ produces

$$Su_1 \circ \cdots \circ Su_t,$$

which realizes a specific combination of subcovers: paths admitted by earlier subcovers are composed with those of later ones. Because $\circ$ is, in general, noncommutative, the order of $Su_i$ reflects the intended sequencing of interactions encoded by the cover.

## F.2 COMPARING WITH MPNN

For a node $v$, its neighborhood can be described by the element $\mathsf{Sieve}(v, 1)$. Consequently, $\mathsf{SNN}_o(\alpha, (0, 1))$ and $\mathsf{SNN}_o(\alpha, (1, 0))$ correspond to the adjacency matrix, signifying their utilization of neighborhoods for message passing. This is equivalent to MPNNs. Hence, SNN can be considered as a generalization of MPNNs. In the following example, two graphs are considered that MPNN can not distinguish, yet SNN can. This example illustrates how a shift in perspective, resulting from a change in cover, reveals the topological properties of graphs.

**Example F.2.1.** *The graphs in Figure 3 are not distinguishable by MPNN (Sato, 2020) because they are locally the same. Applying $\mathsf{SNN}_o(\alpha, (1, 1))$, a level of version $\alpha$ of SNN that is slightly more*

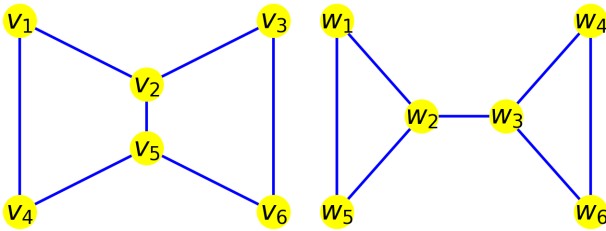

Figure 3: The graph $G$, the left one, and $H$, the right one, are not distinguishable by MPNN

*potent than MPNN, we get the following symmetric matrices $X$ and $Y$ for $G$ and $H$ respectively as the outputs of the model for these graphs.*

$$X = \begin{pmatrix} 2 & 2 & 1 & 2 & 2 & 0 \\ 2 & 3 & 2 & 2 & 2 & 2 \\ 1 & 2 & 2 & 0 & 2 & 2 \\ 2 & 2 & 0 & 2 & 2 & 1 \\ 2 & 2 & 2 & 2 & 3 & 2 \\ 0 & 2 & 2 & 1 & 2 & 2 \end{pmatrix} \quad Y = \begin{pmatrix} 2 & 3 & 1 & 0 & 3 & 0 \\ 3 & 3 & 2 & 1 & 3 & 1 \\ 1 & 2 & 3 & 3 & 1 & 3 \\ 0 & 1 & 3 & 2 & 0 & 3 \\ 3 & 3 & 1 & 0 & 2 & 0 \\ 0 & 1 & 3 & 3 & 0 & 2 \end{pmatrix}$$

Table 3: Dataset statistics and runtime (in seconds) for constructing SNN-transformed graphs. Reported times correspond to building the cover operator before applying Label Propagation.

| Dataset | #Nodes | #Edges | #Classes | Runtime $(\mathsf{SNN}(\beta, (1,1)))$ | Runtime $(\mathsf{SNN}(\beta, (1,1,1)))$ |
|---------|--------|--------|----------|------------------|------------------|
| Cora | 2,708 | 10,556 | 7 | 0.0525 | 0.0758 |
| CiteSeer | 3,327 | 9,104 | 6 | 0.0189 | 0.0319 |
| PubMed | 19,717 | 88,648 | 3 | 0.8055 | 1.3902 |
| ogbn-arxiv | 169,343 | 1,166,243 | 40 | 6.75 | 9.38 |

*The entry $ij$ in these matrices corresponds to the count of paths between nodes $v_i$ and $v_j$ in $\mathsf{CoSieve}(v_i, 1) \bullet \mathsf{Sieve}(v_j, 1)$ and $w_i$ and $w_j$ in $\mathsf{CoSieve}(w_i, 1) \bullet \mathsf{Sieve}(w_j, 1)$. The disparity between these matrices highlights the differences between the graphs. This dissimilarity becomes more apparent when applying the set function $\mathsf{Var}$, while $\mathsf{Sum}$ and $\mathsf{Mean}$ yield identical values. When $\mathsf{SNN}_o(\alpha, (1, 2))$, a more complex level of $\mathsf{SNN}$, is applied, we obtain the following nonsymmetric matrices, denoted as $Z$ and $W$, for graphs $G$ and $H$. Applying all three set functions results in distinct outputs, further emphasizing the dissimilarity between the graphs.*

$$
Z = \begin{pmatrix} 2 & 4 & 2 & 4 & 4 & 3 \\ 5 & 3 & 5 & 4 & 6 & 4 \\ 2 & 4 & 2 & 3 & 4 & 4 \\ 4 & 4 & 3 & 2 & 4 & 2 \\ 4 & 6 & 4 & 5 & 3 & 5 \\ 3 & 4 & 4 & 2 & 4 & 2 \end{pmatrix} \quad W = \begin{pmatrix} 2 & 3 & 3 & 1 & 3 & 1 \\ 4 & 3 & 4 & 2 & 4 & 2 \\ 2 & 4 & 3 & 4 & 2 & 4 \\ 1 & 3 & 3 & 2 & 1 & 3 \\ 3 & 3 & 3 & 1 & 2 & 1 \\ 1 & 3 & 3 & 3 & 1 & 2 \end{pmatrix}
$$

### F.3 COMPLEXITY

SNN is applied *once* as a preprocessing step to convert each input graph (or a dataset of graphs) into its transformed counterpart; it is not used during training.

Let $G = (V, E)$ with $|V| = n$ and $|E| = m$. From Eq. equation 3, $\mathsf{Image}(v, k)$ is obtained by $k$ iterations of adjacency-based additions/multiplications. The cost depends on the configuration:

- $\mathsf{SNN}(\beta, (1, \ldots, 1))$. In this case $\mathsf{Image}(v, k)$ can be read off directly from the adjacency matrix (no matrix–matrix products), so each $Su_i$ equals the adjacency matrix. Hence computing $S^{(\beta)}$ is $\mathcal{O}(mn)$.

- $\mathsf{SNN}(\alpha, (l, k))$ or $\mathsf{SNN}(\beta, (l_1, \ldots, l_t))$ with $k > 1$ **or some** $l_{i_0} > 1$. These require matrix-based compositions; computing $\mathsf{Image}(v, k)$ for a single node costs $\mathcal{O}(mn)$, yielding $\mathcal{O}(mn^2)$ over all nodes.

Since SNN runs only once to produce the transformed graphs, its runtime is incurred offline and does not affect the training-time complexity of downstream GNNs.

### F.4 ALGORITHM

---

**Algorithm 1** Computing $\mathsf{Image}(v, k)$

---

1: **Input:** node $v$, integer $k$
2: **Output:** $\mathsf{Image}(v, k)$
3: **Initialization:** $N_0(v) = \{v\}$, $\mathsf{Image}(v, 0) = $ *Zero matrix*
4: **for** $i = 1, \cdots, k$ **do**
5: $\quad N_i(v) = \bigcup_{u \in N_{i-1}(v)} N(u) - \bigcup_{j=0}^{i-1} N_j(v)$
6: $\quad M_i(v) = \{w \to u : wu \in E, w \in N_i(v), u \in N_{i-1}(v)\}$
7: $\quad \mathsf{Tr}(D_i(v)) = \sum_{e \in M_i(v)} \mathsf{Tr}(e) = $ The adjacency matrix of directed subgraph $M_i(v)$
8: $\quad \mathsf{Image}(v, i) = \mathsf{Tr}(D_i(v)) \circ \mathsf{Image}(v, i-1)$
9: **end for**
10: **Return:** Final result

---

**Algorithm 2** Computing $\mathsf{ColImage}(v, k)$

---

1: **Input:** $\mathsf{Image}(v, k)$
2: **Output:** $\mathsf{ColImage}(v, k)$
3: $\mathsf{ColImage}(\mathsf{v}, \mathsf{k}) = $ Transpose of $\mathsf{Image}(\mathsf{v}, \mathsf{k})$
4: **Return:** Final result

---

**Algorithm 3** Computing $\mathsf{SNN}(\alpha, (l, k))$

---

1: **Input:** $\mathsf{Image}(v, k)$ and $\mathsf{ColImage}(v, l)$ for all $v \in V$
2: **Output:** $\mathsf{SNN}(\alpha, (l, k))$
3: **Initialization:** $\mathsf{SNN}(\alpha, (l, k)) = $ *Zero matrix*
4: **for** $v_i \in V$ **do**
5: $\quad$ **for** $v_j \in V$ **do**
6: $\quad\quad A = \mathsf{ColImage}(v_i, l) \circ \mathsf{Image}(v_j, k)$
7: $\quad\quad r = \mathsf{ColImage}(v_i, l)[i, :].sum()$, summation of $i-$th row
8: $\quad\quad c = \mathsf{Image}(v_j, k)[:, j].sum()$, summation of $j-$th column
9: $\quad\quad \mathsf{SNN}(\alpha, (l, k))_{i,j} = \frac{A_{i,j}}{r \cdot c}$
10: $\quad$ **end for**
11: **end for**
12: **Return:** Final result

---

**Algorithm 4** Computing $\mathsf{SNN}(\beta, (l, k))$

---

1: **Input:** $\mathsf{Image}(v, k)$ and $\mathsf{ColImage}(v, l)$ for all $v \in V$
2: **Output:** $\mathsf{SNN}(\beta, (l, k))$
3: $Su_1 = \sum_{v \in V} \mathsf{ColImage}(v, l)$
4: $Su_2 = \sum_{v \in V} \mathsf{Image}(v, k)$
5: $\mathsf{SNN}(\beta, (l, k)) = Su_1 \circ Su_2$
6: **Return:** Final result

---

## G SPECIAL SUBMONOIDS

### G.1 THE SUBMONOID GENERATED BY NEIGHBORHOODS

The cover of neighborhoods, as a subset of $\mathsf{Mod}(G)$, generates a submonoid. To formalize this, let $\mathsf{Neigh}(G) \subseteq \mathsf{Mod}(G)$ and $\mathsf{Adj}(G) \subseteq \mathsf{Mom}(G)$ denote the submonoids generated by the cover of neighborhoods and its matrix transformation, respectively. The following theorems illustrate how these submonoids provide an algebraic characterization of a graph. It is straightforward to verify that for a graph isomorphism $f : G \to H$, the mappings $\mathsf{Mod}(f)$ and $\mathsf{Mom}(f)$ send elements of $\mathsf{Neigh}(G)$ and $\mathsf{Adj}(G)$ to elements of $\mathsf{Neigh}(H)$ and $\mathsf{Adj}(H)$, respectively. Thus, as a consequence of Theorem 3.1.2, we have:

**Theorem G.1.1.** *Every graph isomorphism $f : G \to H$ induces monoidal isomorphisms $\mathsf{Neigh}(f) :$ $\mathsf{Neigh}(G) \longrightarrow \mathsf{Neigh}(H)$ and $\mathsf{Adj}(f) : \mathsf{Adj}(G) \to \mathsf{Adj}(H)$ such that the following diagram is commutative, where $\iota$ represents the inclusions.*

$$
\begin{array}{ccccc}
\mathsf{Neigh}(G) & \xrightarrow{\mathsf{Tr}_G} & \mathsf{Adj}(G) & \xrightarrow{\iota} & \mathsf{Mat}_{|V_G|}(\mathbb{R}) \\
\downarrow{\scriptstyle \mathsf{Neigh}(f)} & & \downarrow{\scriptstyle \mathsf{Adj}(f)} & & \downarrow{\scriptstyle \mathsf{CO}(f)} \\
\mathsf{Neigh}(H) & \xrightarrow[\mathsf{Tr}_H]{} & \mathsf{Adj}(H) & \xrightarrow[\iota]{} & \mathsf{Mat}_{|V_H|}(\mathbb{R})
\end{array}
\tag{4}
$$

The converse of this theorem can be stated as follows:

**Theorem G.1.2.** *Suppose $G$ and $H$ are two graphs with $|V_G| = |V_H| = n$, and $f : \mathsf{Mat}_n(\mathbb{R}) \to$ $\mathsf{Mat}_n(\mathbb{R})$ is a Change-of-Order mapping. If the restriction of $f$ to $\mathsf{Adj}(G)$ yields an isomorphism to $\mathsf{Adj}(H)$, then $G$ and $H$ are isomorphic.*

Consequently, the horizontal homomorphisms in Diagram 4 can serve as an algebraic description of the graph. It demonstrates that the monoidal elements resulting from interactions between neighborhoods encapsulate richer information about the graph's topology. This suggests that the coverage of neighborhoods can be further enhanced by incorporating additional elements from $\mathsf{Neigh}(G)$.

### G.2 THE SUBMONOID GENERATED BY SIEVES

The submonoid generated by the cover of Sieves fully determines the graph, as stated in the following two theorems. Let $\mathsf{Si}(G) \subseteq \mathsf{Mod}(G)$ and $\mathsf{Im}(G) \subseteq \mathsf{Mom}(G)$ denote the submonoids generated by the cover of sieves and its matrix transformation, respectively. As a direct consequence of Theorems 4.1.1 and 3.1.2, we have:

**Theorem G.2.1.** *Every graph isomorphism $f : G \to H$ induces monoidal isomorphisms $\mathsf{Si}(f) :$ $\mathsf{Si}(G) \longrightarrow \mathsf{Si}(H)$ and $\mathsf{Im}(f) : \mathsf{Im}(G) \to \mathsf{Im}(H)$ such that the Diagram 5 is commutative, where $\iota$ represents the inclusions.*

$$
\begin{array}{ccccc}
\mathsf{Si}(G) & \xrightarrow{\mathsf{Tr}_G} & \mathsf{Im}(G) & \xrightarrow{\iota} & \mathsf{Mat}_{|V_G|}(\mathbb{R}) \\
\downarrow{\scriptstyle \mathsf{Si}(f)} & & \downarrow{\scriptstyle \mathsf{Im}(f)} & & \downarrow{\scriptstyle \mathsf{CO}(f)} \\
\mathsf{Si}(H) & \xrightarrow[\mathsf{Tr}_H]{} & \mathsf{Im}(H) & \xrightarrow[\iota]{} & \mathsf{Mat}_{|V_H|}(\mathbb{R})
\end{array}
\tag{5}
$$

The converse of the above theorem can be stated as follows:

**Theorem G.2.2.** *Suppose $G$ and $H$ are two graphs with $|V_G| = |V_H| = n$, and $f : \mathsf{Mat}_n(\mathbb{R}) \to$ $\mathsf{Mat}_n(\mathbb{R})$ is a Change-of-Order mapping. If the restriction of $f$ to $\mathsf{Im}(G)$ yields an isomorphism to $\mathsf{Im}(H)$, then $G$ and $H$ are isomorphic.*

The horizontal morphisms in Diagram 5 provide a unique, up-to-isomorphism algebraic characterization of a graph. This can be served as the basis for the significant performance of the model SNN in the graph isomorphism task, as demonstrated in the experimental section.

# H PROOF OF THEOREMS

## H.1 PROOF OF THEOREM 2.1.3

*Proof.* Since Rep is surjective, it suffices to demonstrate that Rep is also injective, meaning that if $\mathsf{Rep}(D) = \mathsf{Rep}(D')$, then $D = D'$. According to the matrix representation definition, $\leq_D = \leq_{D'}$. For an edge $v_i \xrightarrow{e} v_j$ in $D$, it implies $v_i \leq_D v_j$, and consequently, $v_i \leq_{D'} v_j$. Suppose $v_i \xrightarrow{e} v_j$ is not a directed edge in $D'$. In that case, there must be a path in $D'$ traversing a node $v_k$ different from $v_i$ and $v_j$. This implies $v_i \leq_{D'} v_k$ and $v_k \leq_{D'} v_j$, and consequently, $v_i \leq_D v_k$ and $v_k \leq_D v_j$. Thus, there is a path in $D$ from $v_i$ to $v_j$ traversing $v_k$. However, this path is distinct from

$v_i \xrightarrow{\ e\ } v_j$ , contradicting the definition of directed subgraphs. Therefore, $e$ is a directed edge in $D'$. Similarly, we can demonstrate that every edge in $D'$ also belongs to $D$ with the same direction. Thus, $D = D'$. □

## H.2 PROOF OF THEOREM 2.2.1

*Proof.* The empty graph is its identity element, and the associativity of • comes from the associativity of the composition of paths. The non-commutativity is explained in Example 2.2.2. □

## H.3 PROOF OF THEOREM 2.2.5

*Proof.* Since directed subgraphs, together with the operation • generate the monoid $\mathsf{Mod}(G)$, we just need to show that every directed subgraph can be formed by its directed edges using the operation •. We will prove this by induction based on the number of edges. Let $D$ be a directed subgraph of $G$. There is nothing to prove if $D$ has just one directed edge. Suppose the number of edges in $D$ is $m$, and the statement is true for every directed subgraph with edges less than $m$; Our task is to show that the statement holds for $D$ as well.

Let $V_D$ be the set of nodes of $D$. Since $\leq_D$ is transitive, $(V_D, \leq_D)$ can be seen as a partially ordered set, implying the existence of maximal elements. A node is considered maximal if it is not the starting point of any path. Now, let $v$ be a maximal node; we choose a directed edge $w \xrightarrow{\ e\ } v$ in $D$ and remove it. The following three situations may occur:

1) producing one directed subgraph $D'$: $D$ and $D' \bigoplus e$ have the same directed edges. Since $v$ is maximal, the paths of $D$ that pass $e$ have this directed edge as their terminal edge. Then
$$\mathsf{Paths}(D) = \mathsf{Paths}(D') \star e$$
This follows $D = D' \bullet e$. Based on the assumption, $D'$ can be created by its edges. Then, the statement is true for $D$.

2) producing two components where one of them is an isolated node, and the other one is a directed subgraph $D'$: in this case, we first remove the isolated node and then, similar to the first case, we conclude that the statement is true for $D$.

3) producing two directed subgraphs $D'$ and $D''$ where $w \in D'$ and $v \in D''$: obviously $D$ and $D' \bigoplus e \bigoplus D''$ have the same directed edges. With an argument similar to the first part, the maximality of $v$ implies
$$\mathsf{Paths}(D) = \mathsf{Paths}(D') \star \{e\} \star \mathsf{Paths}(D'')$$
and then $D = D' \bullet e \bullet D''$. Now, by the assumption that $D'$ and $D''$ can be created by their edges, the statement is true for $D$.

□

## H.4 PROOF OF THEOREM 2.3.1

*Proof.* Since the summation and multiplication of matrices are associative, the operation ○ is associative. The zero matrix is the identity element of $\mathsf{Mat}_n(\mathbb{R})$ with respect to ○. □

## H.5 PROOF OF THEOREM 2.3.3

To define a monoidal homomorphism between the monoids $(\mathsf{Mod}(G), \bullet)$ and $(\mathsf{Mom}(G), \circ)$ in such a way that it is an extension of the morphism $\mathsf{Rep}$, we first prove the following theorem which gives a good explanation of the monoidal operation ○.

**Theorem H.5.1.** *For $A_1, A_2, \cdots, A_k \in \mathsf{Mat}_n(\mathbb{R})$ with $k \in \mathbb{N}$ we have:*

$$A_1 \circ A_2 \circ \cdots \circ A_k = \sum_{i=1}^{k} A_i + \sum_{\sigma \in O(k,2)} A_{\sigma_1} A_{\sigma_2} + \cdots + \sum_{\sigma \in O(k,j)} A_{\sigma_1} \cdots A_{\sigma_j} + \cdots + A_1 A_2 \cdots A_k$$

*where $O(k, i)$ is the set of all strictly monotonically increasing sequences of $i$ numbers of $\{1, \cdots, k\}$*

*Proof.* We prove the statement by induction on $k$. For $k = 2$, there is nothing to prove, which is clear from the definition. Let the statement be true for $k$; We will show it is true for $k + 1$. The associativity of $\circ$ and the induction hypothesis imply:

$$A_1 \circ A_2 \circ \cdots \circ A_k \circ A_{k+1} = (A_1 \circ A_2 \circ \cdots \circ A_k) \circ A_{k+1} =$$

$$(A_1 \circ A_2 \circ \cdots \circ A_k) + A_{k+1} + (A_1 \circ A_2 \circ \cdots \circ A_k)A_{k+1} =$$

$$\sum_{i=1}^{k} A_i + \cdots + \sum_{\sigma \in O(k,j)} A_{\sigma_1} \cdots A_{\sigma_j} + \cdots + A_1 A_2 \cdots A_k +$$

$$A_{k+1} +$$

$$(\sum_{i=1}^{k} A_i + \cdots + \sum_{\sigma \in O(k,j)} A_{\sigma_1} \cdots A_{\sigma_j} + \cdots + A_1 \cdots A_k)A_{k+1} =$$

$$= \sum_{i=1}^{k+1} A_i + (\sum_{i=1}^{k} A_i A_{k+1} + \sum_{\sigma \in O(k,2)} A_{\sigma_1} A_{\sigma_2}) + \cdots +$$

$$(\sum_{\sigma \in O(k,j-1)} A_{\sigma_1} \cdots A_{\sigma_{j-1}} A_{k+1} + \sum_{\sigma \in O(k,j)} A_{\sigma_1} \cdots A_{\sigma_j}) +$$

$$\cdots + A_1 \cdots A_k A_{k+1} =$$

$$\sum_{i=1}^{k+1} A_i + \sum_{\sigma \in O(k+1,2)} A_{\sigma_1} A_{\sigma_2} + \cdots + \sum_{\sigma \in O(k+1,j)} A_{\sigma_1} \cdots A_{\sigma_j} +$$

$$\cdots + A_1 A_2 \cdots A_k A_{k+1}$$

Therefore the statement is true for $k + 1$. $\qquad\square$

Now, we prove Theorem 2.3.3.

*Proof.* Considering that $S = \mathsf{Paths}(D_1) \star \cdots \star \mathsf{Paths}(D_k)$, let $p = p_0 p_1 \cdots p_m \in S$ be a path from $v_i$ to $v_j$ that is obtained by composition of subpaths $p_0 \in \mathsf{Paths}(D_{i_0}), \cdots, p_m \in \mathsf{Paths}(D_{i_m})$ and $1 \le i_0 \lneq \cdots \lneq i_m \le k$. The number of all such paths from $v_i$ to $v_j$ equals the $ij$ entry of the matrix $(A_{i_0} \cdots A_{i_m})$ that is a summand of $A$ as explained in Theorem H.5.1. So the number of all paths from $v_i$ to $v_j$ in $S$ equals the $ij$ entry of $A$. Therefore, the definition of $\mathsf{Tr}$ just depends on $S$ and is independent of the choice of $D_i$s. Then $\mathsf{Tr}$ is well-defined. Based on the definition, $\mathsf{Tr}$ is a monoidal homomorphism.

Suppose $B \in \mathsf{Mom}(G)$, then there are some matrix representations $B_1, \cdots, B_l$ in $\mathsf{MatRep}(G)$ such that $B = B_1 \circ \cdots \circ B_l$. Since $\mathsf{Rep}$ is an isomorphism, there exist some directed subgraphs $C_1, \cdots, C_l$ such that $\mathsf{Rep}(C_i) = B_i$. Now, by choosing $C = C_1 \bullet \cdots \bullet C_l$, we obtain $\mathsf{Tr}(C) = B$, establishing that $\mathsf{Tr}$ is surjective. $\qquad\square$

## H.6   PROOF OF PROPOSITION 3.1.1

*Proof.* As we explained, $f$ changes the order of rows and columns. Thus, it preserves element-wise and matrix multiplications. Since $f$ is also linear, we have

$$\begin{aligned}
f(A \circ B) &= f(A + B + AB) \\
&= f(A) + f(B) + f(AB) \\
&= f(A) + f(B) + f(A)f(B) \\
&= f(A) \circ f(B)
\end{aligned}$$

and then $f$ preserves the operation $\circ$ and this property establishes $f$ as a monoidal isomorphism. $\quad\square$

## H.7 PROOF OF THEOREM 3.1.2

*Proof.* Since $f$ is a change in the order, it induces bijections $\mathsf{DirSub}(f)$ and $\mathsf{MatRep}(f)$ such that Diagram 6 commutes.

$$
\begin{array}{ccc}
\mathsf{DirSub}(G) & \xrightarrow{\ \mathsf{Rep}\ } & \mathsf{MatRep}(G) \\
{\scriptstyle \mathsf{DirSub}(f)}\downarrow & & \downarrow{\scriptstyle \mathsf{MatRep}(f)} \\
\mathsf{DirSub}(H) & \xrightarrow[\ \mathsf{Rep}\ ]{} & \mathsf{MatRep}(H)
\end{array}
\tag{6}
$$

Also, $f$ induces monoidal isomorphism $\mathsf{SMult}(f) : \mathsf{SMult}(G) \to \mathsf{SMult}(H)$ that sends $(M, S) \mapsto (f(M), f(S))$. According to the commutativity of the squares in Diagram 7, isomorphisms $\mathsf{Mod}(f) : \mathsf{Mod}(G) \to \mathsf{Mod}(H)$ and $\mathsf{Mom}(f) : \mathsf{Mom}(G) \to \mathsf{Mom}(H)$ can be obtained by restricting $\mathsf{SMult}(f)$ to $\mathsf{Mod}(G)$ and $\mathsf{CO}(f)$ to $\mathsf{Mom}(G)$.

$$
\begin{array}{ccccc}
\mathsf{DirSub}(G) \xrightarrow{\mathsf{DirSub}(f)} \mathsf{DirSub}(H) & & & \mathsf{MatRep}(G) \xrightarrow{\mathsf{MatRep}(f)} \mathsf{MatRep}(H) \\
\downarrow \qquad\qquad \downarrow & & & \downarrow \qquad\qquad\qquad \downarrow \\
\mathsf{SMult}(G) \xrightarrow[\mathsf{SMult}(f)]{} \mathsf{SMult}(H) & & & \mathsf{Mat}_{|V_G|}(\mathbb{R}) \xrightarrow[\mathsf{CO}(f)]{} \mathsf{Mat}_{|V_H|}(\mathbb{R})
\end{array}
\tag{7}
$$

The commutativity of the right square in Diagram 1 directly follows from the definition of $\mathsf{Mom}(f)$. As illustrated in Diagram 6, the left square in Diagram 1 is shown to be commutative for the generators of monoids, establishing the commutativity of this square. $\qquad\square$

## H.8 PROOF OF THEOREM 3.1.3

*Proof.* We begin by demonstrating that $f$ establishes a one-to-one correspondence between the edges of $G$ and $H$. It is evident that a matrix with a single non-zero entry in either $\mathsf{Mom}(G)$ or $\mathsf{Mom}(H)$ corresponds to a matrix transformation of an element in $\mathsf{Mod}(G)$ or $\mathsf{Mod}(H)$, respectively, each representing a single directed edge.

For an edge $v_i \text{———} v_j$ in $G$, let $e$ be the directed edge $v_i \to v_j \in \mathsf{Mod}(G)$; then $A = \mathsf{Tr}_G(e)$ has one non-zero entry, and since $f$ is a linear isomorphism, $f(A)$ has one non-zero entry, and, based on the assumption, it belongs to $\mathsf{Mom}(H)$. So $f(A)$ is a matrix transformation of a directed edge $c : u_k \to u_l$ in $\mathsf{Mod}(H)$. Similarly, let $B \in \mathsf{Mom}(G)$ be the matrix transformation of $e' : v_j \to v_i$ and then $f(B) \in \mathsf{Mom}(H)$ is a matrix transformation of some directed edge $c' : u_{l'} \to u_{k'}$ in $\mathsf{Mod}(H)$. Since $e$ can be followed by $e'$, $e \bullet e'$ has three paths. This implies $\mathsf{Tr}_G(e \bullet e')$ has three non-zero entries. On the other hand, $\mathsf{Tr}_G(e \bullet e') = \mathsf{Tr}_G(e) \circ \mathsf{Tr}_G(e') = A \circ B = A + B + AB$; then $AB \neq 0$ and consequently $f(A)f(B) = f(AB) \neq 0$. The equation

$$
\begin{aligned}
\mathsf{Tr}_H(c \bullet c') &= \mathsf{Tr}_H(c) \circ \mathsf{Tr}_H(c') \\
&= f(A) \circ f(B) \\
&= f(A) + f(B) + f(A)f(B)
\end{aligned}
$$

says that the matrix transformation corresponding to $c \bullet c'$ has three non-zero entries and so $c \bullet c'$ contains three paths. Then $c$ must be followed by $c'$ and this yields $u_l = u_{l'}$. Similarly, $u_k = u_{k'}$ can be shown. Therefore, $f$ gives a one-to-one mapping between the edges of $G$ and $H$.

To prove the correspondence between the nodes of two graphs, let $v_x$ be a node in $G$, connected to $v_i$ in which $j \neq x$ and $C$ and $f(C)$ be the matrix transformations of $a : v_i \to v_x \in \mathsf{Mod}(G)$ and $b : u_y \to u_z \in \mathsf{Mod}(H)$, respectively. Since $e'$ is followed by $a$ in $\mathsf{Mod}(G)$, with the same reasoning as above, $c'$ must be followed by $b$ in $\mathsf{Mod}(H)$ and this means $u_k = u_y$. So $f$ also gives a one-to-one mapping between nodes of graphs compatible with edges. Then, $G$ and $H$ are isomorphic. $\qquad\square$

## H.9 Proof of Theorem 3.2.2

The role of neighborhoods in MPNN is like a sink such that messages move to the center of the sink. For a node $v_k$ with neighborhood $N_k$ containing $v_{k_1}, v_{k_2}, \cdots, v_{k_m}$, we depict this sink in Figure 4 by denoting directed edge from $v_{k_i}$ to $v_k$ by $e_i : v_{k_i} \to v_k$. This sink can be considered as a directed

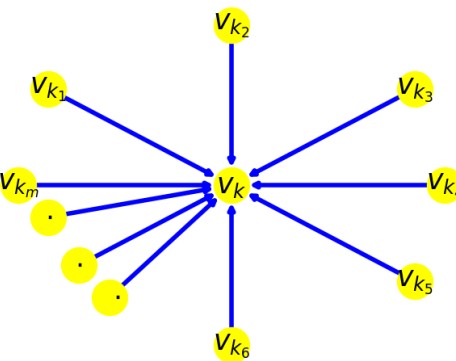

Figure 4: Visualizing a neighborhood by representing it as a directed subgraph

subgraph. As an element of $\mathsf{Mod}(G)$, it can be represented as follows:

$$S_k = e_1 \bullet e_2 \bullet \cdots \bullet e_m$$

Since the directed edges $e_i$ and $e_j$ appearing in $S_k$ are not composable, we observe $e_i \bullet e_j = e_j \bullet e_i$, rendering the order in $S_k$ unimportant. The cover obtained by $S_k$s is exactly the cover of the neighborhoods. Let $T_k = \mathsf{Tr}(S_k)$ and $A_i = \mathsf{Tr}(e_i)$. Thus $A_i$ has 1 in the entry $k_i k$ and 0 for all other entries. The matrix transformation of $e_i \bullet e_j$ has just two non-zero entries and $\mathsf{Tr}(e_i \bullet e_j) = A_i + A_j + A_i A_j$. Then $A_i A_j = 0$ for $1 \leq i \leq m$ and $1 \leq j \leq m$. Theorem H.5.1 implies

$$T_k = \mathsf{Tr}(S_k) = A_1 \circ A_2 \circ \cdots \circ A_m$$
$$= A_1 + A_2 + \cdots + A_m$$

As a result, the column $k$ of $T_k$ aligns with the column $k$ of the adjacency matrix of graph $G$, while the remaining columns are filled with zeros. Transforming the cover $\{S_k\}$ yields a collection of $|V|$ matrices, each containing a single column from the adjacency matrix. In the GGNN framework, summation is an allowed operation, enabling the construction of the adjacency matrix by performing the summation on this matrix collection. Hence, neighborhoods can function as a cover within the framework of GGNN, with the adjacency matrix serving as an interpretation of this cover.

## H.10 Proof of Theorem 4.1.1

*Proof.* Since the definition of sets $M_i(v)$s is based on the neighborhoods, for a graph isomorphism $f : G \to H$, $f(M_i(v)) = M_i(f(v))$. This follows $\mathsf{Mod}(f)(D_i(v)) = D_i(f(v))$. Since $\mathsf{Mod}(f)$ is a monoidal homomorphism, we get:

$$\mathsf{Mod}(f)(\mathsf{Sieve}(v, k)) = \mathsf{Mod}(f)(D_k(v) \bullet \cdots \bullet D_0(v))$$
$$= \mathsf{Mod}(f)(D_k(v)) \bullet \cdots \bullet \mathsf{Mod}(f)(D_0(v))$$
$$= D_k(f(v)) \bullet \cdots \bullet D_0(f(v))$$
$$= \mathsf{Sieve}(f(v), k)$$

Based on Theorem 3.1.2, $\mathsf{Mom}(f)(\mathsf{Image}(v, k)) = \mathsf{Image}(f(v), k)$. $\qquad\square$

## H.11 Proof of Theorem 4.2.1

*Proof.* Since the cover of sieves is invariant and $\mathsf{CO}(f)$ preserves the rest of the computations in the algorithm, SNN is invariant. $\qquad\square$

## H.12 Proof of Theorem G.1.2

*Proof.* Let $\mathsf{Adj}(v)$ denote the matrix representation of the neighborhood of a node $v \in G$. As demonstrated, this matrix contains exactly one non-zero column. The mapping $f$ is a Change-of-Order mapping, which transforms $\mathsf{Adj}(v)$ into a matrix with a single non-zero column, where all non-zero entries are equal to $1$.

An element of $\mathsf{Adj}(H)$ that is not a matrix transformation of any element in the cover of the neighborhood will have two or more non-zero columns. Consequently, for $f(\mathsf{Adj}(v)) \in \mathsf{Adj}(H)$, there exists a node $u \in H$ such that $f(\mathsf{Adj}(v)) = \mathsf{Adj}(u)$.

This establishes a one-to-one correspondence between $V_G$ and $V_H$, as $f$ is an isomorphism. Now, let $v_i \text{——} v_j$ represent an edge in $G$, with $f(\mathsf{Adj}(v_i)) = \mathsf{Adj}(u_k)$ and $f(\mathsf{Adj}(v_j)) = \mathsf{Adj}(u_l)$. The entry $ii$ in the matrix $\mathsf{Adj}(v_j) \circ \mathsf{Adj}(v_i)$ equals $1$.

Since $f$ is a Change-of-Order mapping, the matrix $f(\mathsf{Adj}(v_j) \circ \mathsf{Adj}(v_i)) = \mathsf{Adj}(u_l) \circ \mathsf{Adj}(u_k)$ has a diagonal entry equal to $1$. In this matrix, the only diagonal entry that can be non-zero is the entry $kk$. Similarly, the entry $ll$ in $\mathsf{Adj}(u_k) \circ \mathsf{Adj}(u_l)$ equals $1$. This implies that there is an edge between $u_k$ and $u_l$.

Thus, we establish a one-to-one correspondence between the edges of $G$ and $H$ that is consistent with the mapping of their nodes. This proves that $f$ defines a graph isomorphism between $G$ and $H$. $\square$

