# OpenReview forum: "Grothendieck Graph Neural Networks Framework: An Algebraic Platform for Crafting Topology-Aware GNNs"
_ICLR.cc/2026/Conference — ICLR 2026 Conference Withdrawn Submission_

### Official Review · Reviewer_RYFv · 2025-10-26

**Soundness:** 3
**Presentation:** 3
**Contribution:** 3
**Rating:** 4
**Confidence:** 4

**Summary:**

This paper proposes the Grothendieck Graph Neural Networks (GGNN) framework - an algebraic generalization of neighborhoods for graph representation learning. Rather than relying solely on adjacency-based neighborhoods, GGNN introduces the notion of graph covers, derived from directed subgraph monoids, to construct more expressive message-passing schemes. Through a mapping between subgraph monoids and matrix monoids, GGNN provides a principled algebraic platform that recovers the adjacency matrix as a special case. As an instantiation, the authors introduce the Sieve Neural Network (SNN), inspired by sieves in category theory, which constructs "sieve covers" that capture rich directional and topological relationships. Theoretical results show that these constructions preserve graph isomorphism invariance and characterize graphs algebraically up to isomorphism. Empirically, SNN achieves 0\% failure on challenging graph isomorphism benchmarks (SR, CSL, BREC) and significantly improves topology encoding in label propagation probes on citation and OGB datasets

**Strengths:**

1. This paper establishes a novel algebraic foundation for GNNs, generalizing neighborhoods through monoidal category structures and Grothendieck-like constructions.
2. The introduction of the cover of sieves offers a categorically motivated yet computationally realizable mechanism for topology-aware message passing - a fusion of abstract algebra and neural architectures.
3. The experimental evaluation is diverse and meaningful: from graph isomorphism to topology encoding using parameter-free probes, directly connecting algebraic design to empirical outcomes.
4. The paper is clearly structured, progressing from theoretical definitions to architectural instantiation (SNN) to empirical results.
5. This paper provides a general algebraic framework rather than a single architecture, enabling the design of infinitely many GNN variants grounded in mathematical principles.

**Weaknesses:**

1. **Abstract Algebra Overhead.:** The heavy reliance on categorical formalism may obscure intuition. The paper could include simplified interpretations of monoid operations to make the framework more accessible.
2. **Lack of Ablation and Sensitivity:** The improvement trends in Table 2 suggest cover depth correlates with performance, but no controlled ablation isolates this factor. Quantifying how depth $k$ affects accuracy and runtime would provide design guidance.
3. **Large Graph Classification/Regression Evaluation:** Could the authors clarify whether the proposed GNN can scale to large graph classification/ regression tasks such as ogb benchmarks? If scalability is not yet feasible, could the authors discuss possible approximations that would enable application to such large graphs?
4. **Theorem Completeness and Proof Detail:** While Theorems 2.3.3 and 3.1.3 are formally correct, they rely on assumed surjectivity and Change-of-Order mappings without proof of uniqueness. An explicit counterexample discussion would make the claims more transparent.
5. **Connection to Existing Expressivity Results:** The paper should discuss how GGNN compares in expressivity to 2-WL or 3-WL universal GNNs in formal terms. Currently, the relationship is empirical rather than theoretical.

**Questions:**

1. **Expressivity relative to WL hierarchy:** The paper claims zero failures on benchmarks that defeat 3-4-WL tests, suggesting higher expressivity. Could the authors formally position GGNN/SNN within or beyond the WL hierarchy, perhaps via a bounded equivalence or simulation theorem?
2. **Computational complexity of cover construction:** The algebraic formalism is elegant but potentially expensive. What is the theoretical and empirical complexity (time and space) of constructing sieve covers for large graphs (i.e., ogb graph classification/ regression tasks)? A complexity comparison with standard message passing would clarify scalability.
3. **Theoretical guarantees of completeness:** Theorem 3.1.3 provides a partial converse between isomorphism and monoid equivalence. Under what additional constraints does this converse become full equivalence? This would strengthen the theoretical soundness of GGNN as a graph characterization tool.
4. **Ablation and sensitivity analysis:** Performance on LP probes improves with deeper covers $((1,1,1) > (1,1))$, but there's no analysis of depth sensitivity or runtime trade-offs. Such results would inform practical use.

---

### Official Review · Reviewer_N296 · 2025-10-31

**Soundness:** 4
**Presentation:** 3
**Contribution:** 3
**Rating:** 4
**Confidence:** 4

**Summary:**

In the submitted manuscript, the authors define a multitude of monoids and representation maps that allow them to generalise the cover of neighbourhoods on graphs, which is the common primitive for message passing in Graph Neural Networks (GNNs). This algebraic generalisation allows the authors to propose a more general framework for learning on graphs and in particular to instantiate a particular version of their generalised GNNs, the Sieve Neural Networks (SNNs). The authors finally report strong performance of their SNNs on the SR, CSL and BREC benchmarks as well as the Cora, CiteSeer, PubMed and ogbn-arxiv datasets.

**Strengths:**

- The submitted work is strongly theoretically grounded and presents a principled approach to deep learning on graphs.
- The empirical results on the expressivity benchmarks are impressive.

**Weaknesses:**

- I found it a bit difficult to follow the extensive theoretical discussion in Sections 2, 3 and 4. Of course, this may be as much my fault as the authors'. But I feel that the paper could be edited to be even more accessible. For example, a running example in Section 2.2 following the three-step procedure you outline may be nice. It may furthermore be nice if the authors could expand their discussion of Example 2.3.4, for example, by explicitly showing the obtained matrices.

- It seems to me that the experimental evaluation may be incomplete in a sense (please find more detail in Question 4] below).

**Questions:**

1] To me, it seems that your comparison to existing GNNs in Appendix E is insufficient. A general framework for learning on graphs, as you describe here, could be relevant if it is either shown to include many existing GNNs as special cases to unify their analysis or to give rise to novel GNNs that have not yet been proposed. It seems to me that you do not sufficiently pursue either option. It is unclear to what extent existing GNNs fit into your framework (besides the single case of Feng et al. 2022, that you discuss in Appendix E). It is furthermore unclear to me from your work how your SSNs are significantly different from the large variety of GNNs that have already been proposed. Could you please offer a more comprehensive comparison to existing GNNs?

2] I suppose your SNNs may not be able to capture cycles in graphs, since they exclude edges within neighbourhoods in their "layer" formulation in Line 306. Is this right, and may it be worth discussing in the paper? Empirically, the ability to capture cycle information can be of great relevance in our current benchmark datasets in the literature.

3] I could not easily understand the definition of $S_{ij}^{(\alpha)}$ intuitively in Lines 379-80. What is the node-pair-specific capacity that you refer to here exactly? Which path counts are included in this capacity that are missing in the numerator?

4] Concerning your experiments, I have the following questions:

4.1] It is unclear to me why you do not experiment on the SNN($\alpha, (l,k)$) variant in your experiments in Section 5. Would it not be appropriate to also show results for this second instantiation of your SNNs that you provide?

4.2] I am curious whether you ablated the impact of the features that you define in Line 415 on your results. How crucial are these summary statistics of your operator S on the performance of your SNNs?

4.3] For your results in Table 2, I wonder whether the comparison is fair here. How many test nodes are reached by the different adjacency matrices versus the SNN operators? Could it just be the case that the SNN operators have a higher edge density and are therefore able to propagate the label information further than the adjacency matrix?

5] Minor Comments:

5.1] In Line 85 you state that the node set V is equipped with a fixed ordering. Is this really required by you later on? I don't really see why you would not work on graphs with standard node sets.

5.2] In Proposition 3.1.1 you say that change of order mappings are "compatible with monoidal operation \circ". How is the word "compatible" defined here and is it a rigorous use of this term?

5.3] I think you are discussing the inverse degree matrix in Line 456 and not the degree matrix with these normalisations. Is that right?

---

### Official Review · Reviewer_Mz1x · 2025-11-01

**Soundness:** 3
**Presentation:** 1
**Contribution:** 2
**Rating:** 4
**Confidence:** 3

**Summary:**

The article introduces a principled generalisation of ordinary neighbourhood-based message passing using the concept of covers. The idea is to transform the covers into matrices which can then be directly used with common GNN architectures. As an example, the authors introduce SNNs which reflect the number of paths between nodes along specific directed graphs. In an experimental section the authors validate the expressive power and topology-capturing capabilities of SNNs.

**Strengths:**

1. The article offers a comprehensive theory providing formal proofs and theoretical guarantees on the general GGNN concept.
2. The approach provides an interesting generalization of ordinary GNNs.
3. The experimental evaluation of synthetic datasets shows strong results on the expressive power of SNNs.

**Weaknesses:**

1. The article is hard to follow due to its excessive formalisations. It partly reads like a list of notions, definitions and theorems. I would advise the authors to provide more high-level intuitions and visualisations.
2. I believe the focus of the article is misaligned. The ratio of overly generalised mathematical concepts in sections 2 and 3 to the actual main contribution (i.e. SNN) is too disproportionate. Many theoretical results are not immediately necessary to introduce the much simpler concept of SNNs.
3. While the authors put much effort into the theoretical analysis of the general GGNN concept, they analyse the expressive power of the SNN variants only practically. This creates a large gap between theory and practice. The paper is missing a formal expressiveness theory connecting the algebraic framework to SNNs. I believe this severely undermines the central contribution of the paper.
4. While the authors claim to provide a general framework, it is not clear how one would be able to systematically design a new GNN with it.
5. Some results are quite unnecessary. E.g., I believe Theorem 2.3.1 is trivial and does not qualify as a theorem.

**Questions:**

1. Can you provide a formal characterization of SNN's expressive power?
2. Did you perform any ablation studies where e.g. you use sieves without co-sieves, or analyse the effect of path length?

---

### Official Review · Reviewer_1bJ4 · 2025-11-04

**Soundness:** 2
**Presentation:** 2
**Contribution:** 1
**Rating:** 2
**Confidence:** 4

**Summary:**

This paper proposes an new computational design language named as GGNN, the language provides the consistent mapping between multigraph directed subgraph operation and the matrix operation A + B + AB. The author shows that the operation has graph equivariant property under isomorphism, and construct an example network called Sieve Neural Network by selecting a set of covers as k-hop neighbors and doing a series of A + B + AB operations to get an enriched matrix representation that being the input to a normal GNNs. In general, I think the main story or contribution of the paper is about the connection between matrix operation and the closure space of directed subgraphs. The proposed method is a deterministic preprocessor.

**Strengths:**

The connection is interesting, to show the geometric meaning of the matrix operation A + B + AB, this makes the final transformed matrix "explainable" with higher-order path-composition meaning.

**Weaknesses:**

TBH, this paper looks like from someone with math background that focuses on formal definition and characterize the connection between two set of concepts, while appealing when first read, however the underlying stuff is simple. The author uses math-heavy notation makes the paper less readable which I don't like.

The core part of the paper is try to give geometric/graph composition meaning to the operation A + B + AB. Nevertheless, one can easily find that the operation space is not enlarged at all. It basically has the same core mathematical operation as PPGN, which relies on matrix multiplication AB. That is saying, the designed stuff does not have higher expressivity than PPGN, which is bounded by 2-FWL. The author claims "this paper introduces the GGNN framework, enabling the systematic design of expressive GNNs", which is not true as the resulting stuff is not expressive --- it has heavy expressivity bound. One can argue that you can enlarge the set of starting subgraphs/covers as much as you want, which sounds chicken-and-egg problem.

In general, I personally don't like the paper, nevertheless I acknowledge the connection between the matrix operation space and graph composition space is kind of interesting. The main reason is that I don't see any meaningful implication of the "proposed framework/language", it's just a story which does not change the core operation: A + B + AB.

**Questions:**

Not much.

---

### Note · Authors · 2025-11-20

I have read and agree with the venue's withdrawal policy on behalf of myself and my co-authors.